# Substantial Impact of Later-Life Depression Among Community Older Adults on the Family Caregivers’ Burden in the Home Care Setting of Chiang Mai, Northern Thailand

**DOI:** 10.3390/medicina61010050

**Published:** 2025-01-01

**Authors:** Keisuke Shimizu, Myo Nyein Aung, Saiyud Moolphate, Thin Nyein Nyein Aung, Yuka Koyanagi, Siripen Supakankunti, Motoyuki Yuasa

**Affiliations:** 1Department of Global Health Research, Graduate School of Medicine, Juntendo University, Tokyo 113-8421, Japan; k.shimizu.lq@juntendo.ac.jp (K.S.); moyuasa@juntendo.ac.jp (M.Y.); 2Faculty of International Liberal Arts, Juntendo University, Tokyo 113-8421, Japan; 3Advanced Research Institute for Health Sciences, Juntendo University, Tokyo 113-8421, Japan; 4Department of Public Health, Faculty of Science and Technology, Chiang Mai Rajabhat University, Chiang Mai 50300, Thailand; saiyud_m@cmru.ac.th; 5Department of Family Medicine, Faculty of Medicine, Chiang Mai University, Chiang Mai 50200, Thailand; drthinnyeinaung@gmail.com; 6Global Health and Chronic Conditions Research Group, Chiang Mai University, Chiang Mai 50200, Thailand; 7Faculty of Health Sciences, Tokyo Ariake University of Medical and Health Sciences, Tokyo 135-0063, Japan; y-koyanagi@juntendo.ac.jp; 8Centre of Excellence for Health Economics, Faculty of Economics, Chulalongkorn University, Bangkok 10330, Thailand; siripen.s@chula.ac.th

**Keywords:** home care, family medicine, Asia, aging, community-integrated intermediary care (CIIC), depression, health-related quality of life, TCTR20190412004

## Abstract

*Background and Objectives*: Depression is a common geriatric problem globally. It is particularly burdensome in low- and middle-income countries, where care for older people mainly relies on the family in the absence of long-term care facilities. This study aimed to assess the level of caregivers’ burden among family caregivers who are taking care of older persons with depression in the home care setting within the communities of Chiang Mai, Northern Thailand. *Materials and Methods*: This cross-sectional study investigated 867 pairs of community-dwelling older adults and their family caregivers in Chiang Mai, Thailand. The depression of older people was screened using the 15-item Geriatric Depression Scale. The family caregivers’ burden and quality of life were measured using the Caregiver Burden Inventory (CBI) and the European Quality of Life (EQ) Five Dimension Five Level scales. The analysis applied was multivariable regression. *Results*: Two-thirds of the family caregivers were female. The mean age was 55.3 ± 13.8 years. The family caregivers caring for older persons with depression experienced significantly higher levels of burden in terms of the CBI total score (Coefficient: 10.60, 95% CI: 8.60, 12.60) and lower quality of life in terms of the EQ visual analogue scale (Coefficient: −5.52, 95% CI: −8.41, −2.62). They were more likely to take sick leave from their jobs (adj. OR 4.00, 95% CI: 1.73, 9.24) and more often to get sick (adjusted OR 7.26, 95% CI: 2.68, 19.64) than the caregivers of older adults without depression. *Conclusions*: Urgent interventions to prevent depression among older adults and systematic support to relieve family caregiver burden are necessary. The measures to relieve family caregiver burden include care capacity building, psychological support, respite care service, financial support, and other health promotion activities.

## 1. Introduction

Depression is one of the most common health problems in the world. The global prevalence of depression is estimated to be 4.4% [1]. The Global Burden of Disease Study (GBDS) 2017 reported depressive disorders are the third leading cause of disability [2]. Patients with depression experience persistent sadness, anxiety, or “empty” moods, and the way they feel, think, and handle daily activities, such as sleeping, eating, or working, are affected. So, this disease has a huge impact on individuals in the aspect of health-related quality of life (QOL), work productivity, and relationships with family, friends, and community [3,4]. In addition to the patients with depression themselves, their caregivers, including family members and friends, have to suffer from a significant burden. Patients with depression often require constant supervision and get worse easily, so caregivers experience not only emotional and physical distress but also loss of income and health insurance [5]. Depression is more common in the older generation than in the younger generation. The prevalence of depression among older adults ranges from 15 to 25%, while it depends on the measures to categorize depression and the studied population [6].

Moreover, this is a situation not only in high-income countries but also in low- and middle-income countries. According to a past meta-analysis, the pooled prevalence of depression among older adults in low- and middle-income countries was as high as 10.5% [7].

Population aging is happening in many of the low- and middle-income countries, so depression in older adults is an emerging public health problem in those countries [8]. Additionally, more than 75% of people in low- and middle-income countries do not have access to mental health treatment, so they may not be able to receive known, effective treatment [9].

Thailand is no exception to the global phenomenon of population aging. The proportion of the population aged 60 and over is estimated to increase from 7% in 2001 to 35.8% in 2050 [10]. The prevalence of geriatric depression in Thailand ranges from 6.5% to 18.5%, depending on the studies [11]. According to a past study, factors associated with depression among community-dwelling Thai older adults include being 75 or older, being single, drinking alcohol daily, having diabetes, experiencing a fall in the past year, having self-rated health as neutral, poor/very poor, and having moderate/severe dependency based on the ADL scoring [12]. Additionally, another research showed that 19.6% of depression cases treated at tertiary hospitals in Thailand were treatment-resistant depression [13]. Therefore, taking measures against depression among older adults is an important issue in Thailand as well.

In many low- and middle-income countries, including Thailand, the family remains the main source of care for older adults. Only some older adults living in urban areas use paid services for long-term care. However, it is getting difficult to maintain this system because family size is shrinking, children are living separately—sometimes far from their parents—and life expectancy after reaching old age has extended [14]. The burden of caring for these family members is causing them to become unwell or take time off from work.

Various studies have shown that the caregivers of patients with cancer, neurological diseases, and mental illnesses experience a great caregiving burden and that caring for these patients can lead to the development of depression among the caregivers themselves [15,16,17,18,19]. Additionally, multiple studies have shown that the caregiving burden of patients with dementia is also high [20,21].

However, the evidence on the caregiving burden of older adults with depression is scarce. One study involving a small number of samples found that the caregiving burden of late-onset depression is as high as that of Alzheimer’s disease, so interventions are needed to reduce the burden of caregivers [22]. Another large observational study suggested that caregivers of adults with depression experienced a greater caregiving burden than caregivers of adults with other chronic illnesses [23]. This study also demonstrated that caregivers of adults with depression experienced lower health-related QOL, higher productivity impairment, and increased health resource utilization. However, this study was conducted in European high-income countries, so whether these results are true for middle- and low-income countries remains to be investigated.

Efforts to reduce the burden on families caring for older adults with depression are important in Thailand. Therefore, the purpose of this study is to explore the impact of caring for older adults with depression on family caregivers’ caregiving burden, health-related QOL, physical condition, and employment.

## 2. Materials and Methods

### 2.1. Study Design

This study is a cross-sectional study conducted to investigate the impact of later-life depression among older adults upon the family caregivers’ burden in Maehia city, Chiang Mai province, Northern Thailand.

### 2.2. Data Collection and Participants

This study was conducted in accordance with the Declaration of Helsinki. The research protocol was approved by the WHO Research Ethics Review Committee (WHO/ERC ID: ERC.0003064) dated 7 March 2019 and the Boromrajonani College of Nursing, Lampang Thailand Ethics Review Committee (E2562/005) dated 4 March 2019. It was registered at the Thailand Clinical Trial Registry under trial registration number TCTR20190412004.

This was an analysis of data from a cluster randomized controlled trial: the community-integrated intermediary care (CIIC) project. This cluster randomized controlled trial aimed to evaluate the effect of a new aging care model consisting of care capacity building for family caregivers, respite care services, and long-term care prevention programs for community-dwelling older adults. A total of 2788 participants were recruited in 12 clusters, with 6 in the intervention group with 1509 older adults and 6 in the control group with 1279 older adults in Chiang Mai province, Thailand. The unit of randomization, cluster, was a village. Participants in the intervention clusters received the CIIC intervention after screening for eligibility. The eligibility criterion for a cluster was a village that had more than 300 older persons over 60 years of age at the time of randomization. The inclusion criteria included care recipients aged 60 years or older and their family caregivers (male or female) taking care of them at home and living in the study site for at least 1 year. Those who did not understand the informed consent, those with cognitive impairments, and those who did not consent were excluded.

The data from the baseline survey of one intervention arm before the launch of the intervention were used in this present study. STATA version 17SE (Stata Corporation, College Station, TX, USA) was used for power analysis.

Following the CIIC research protocol, trained research assistants collected data using interviewer-administered survey questionnaires from August to December 2019 [24]. Maehia city, Chiang Mai province, Northern Thailand, was randomly selected as the intervention arm, and 1509 older adults and 867 primary caregivers were recruited with written informed consent. In this subgroup analysis, we analyzed the data of pairs of Thai older adults and their family caregivers; therefore, a total of 867 pairs were included in this study.

### 2.3. Measures

The structured questionnaire consisted of three parts: (1) sociodemographic characteristics, (2) assessment of the caregiver burden of the family caregivers, and (3) the care recipients’ demands. Sociodemographic characteristics of the family caregivers included age, gender, marital status, education, occupation, being the main income supporter of the family, and the relationship with the care recipient. The health-related items were also explored, including underlying diseases such as hypertension, diabetes, and hyperlipidemia and health behaviors of family caregivers such as smoking, drinking alcohol, and exercise habits. Care recipients’ depression was screened by applying the 15-item Geriatric Depression Scale (GDS). The assessment of family caregiver burden level was calculated using the Caregiver Burden Inventory (CBI) score. The health-related QOL of family caregivers was measured using the EQ-5D-5L questionnaire and the EQ visual analogue scale (EQ-VAS). The outcome measures included the rate of caregivers who missed or quit their jobs, took leave from their jobs, or got ill.

#### 2.3.1. Geriatric Depression Scale (GDS)

The 15-item Geriatric Depression Scale is an internationally validated scale used as an effective depression screening tool. The “Yes” or “No” responses are scored according to scoring instructions, and the total score of 5–15 is classified into “probable depression” and 0–4 into “normal”. Its Thai version has been proven to work well and is used within the older population in Thailand [25].

#### 2.3.2. Caregiver Burden Inventory (CBI)

This is a commonly used internationally validated questionnaire consisting of 24 questions. It measures the impact of burden on caregivers, reflecting various areas of the caregiver’s well-being and function. The questions are divided into five dimensions: (a) time-dependence burden, which describes the burden due to restrictions on the caregiver’s time; (b) developmental burden, which describes the caregiver’s feelings of being “off-time” in their development with respect to their peers; (c) physical burden, which describes caregiver’s feelings of chronic fatigue and damage to physical health; (d) social burden, which describes caregivers’ feelings of role conflict and the impact on interpersonal and social relationships with family and friends; and (e) emotional burden, which describes caregivers’ negative feelings toward their care recipients [26]. There are five items in each dimension except for physical burden, which has four items. A 5-point Likert scale, ranging from 0 (not at all disruptive) to 4 (very disruptive) was used to evaluate each item, with a total score in the 0–96 range [27]. We categorized the total CBI score into two groups: one group with a CBI total score of less than 24 and another group with a score of more than or equal to 24. A total score of 24 or more indicates a need to seek some form of respite care. For each dimension, we categorized the score into two groups: one group with a score of zero (not at all disruptive) and another group with a score of more than or equal to 1 (disruptive). A score of 1 or more indicates some kind of caregiver burden.

#### 2.3.3. EQ-5D-5L and EQ-VAS

This is a generic and internationally validated instrument to measure health-related QOL [28,29]. It is comprised of two parts: a descriptive health classifying system (EQ-5D-5L) and a visual analogue scale (EQ-VAS). EQ-5D-5L measures the QOL in 5 dimensions: mobility, self-care, usual activities, pain/discomfort, and anxiety/depression [30]. Each dimension has five levels (1–5) of severity ranging from “no problem” to “severe impairments/unable to”. A validated Thai version of EQ-5D-5L was already available with high reliability and validity [31]. The EQ-VAS is a measure of overall self-rated health status with a vertical VAS where the endpoints are labeled 100 (best imaginable health state) and 0 (worst imaginable health state).

All of the study instruments were translated into Thai, back-translated into English, and retranslated into Thai by independent language experts following the WHO process of translation and adaptation of instruments. Researchers also reviewed and edited these instruments after pilot testing. Cronbach’s alpha reliability coefficients of the GDS, CBI, and EQ-5D-5L were 0.80, 0.77, and 0.89, respectively.

### 2.4. Data Analysis

Initially, the data were cleaned, the variables were recoded as needed, and then an exploratory analysis was conducted. The sociodemographic characteristics were analyzed using descriptive analysis. Frequency and percentages were used for the categorical variables (gender, marital status, education, occupation, being the main income supporter, and health behaviors such as smoking, alcohol drinking, and exercise habits, as well as the underlying diseases such as hypertension, diabetes mellitus, and hyperlipidemia), and the mean (M) and standard deviation (SD) were used for the continuous variable (age). Bivariate analyses were performed to provide an understanding of the survey data before multivariable analysis and to examine the associations between potential covariates (e.g., demographics) and dependent study variables (e.g., health-related QOL). For the CBI total score, CBI five dimensions, and EQ five dimensions, we analyzed them in two ways. First, we applied Wilcoxon signed-rank tests, treating them as continuous variables. After that, we categorized each score into two groups and analyzed them as categorical variables. The total CBI score was categorized into two groups: one group with a CBI total score of less than 24 and another group with a score of more than or equal to 24. Each dimension of the CBI score was categorized into two groups: one group with a score of 0 and another group with a score of more than or equal to 1. Each dimension of the EQ score was also categorized into two groups: one group with a score of 0 and another group with a score of more than or equal to 1. For the continuous variables, Wilcoxon signed-rank tests were used to determine the significant differences between groups, whereas Fisher’s exact tests were used for the categorical variables. The results were fed into a more robust multivariable analysis.

For multivariable analysis, we treated all CBI and EQ scores as continuous variables and applied multiple regression analysis to determine the association between the exposure variable and health-related QOL and care burden measures, after controlling for covariates, including age, sex, marital status, smoking, alcohol drinking, exercise habits, health status, diabetes mellitus, education, occupation, and being the main income supporter of the family. Coefficient, t-value, *p*-value, and 95% confidence interval were used to determine the association. Logistic regression analysis was applied to determine the association between the population group and the rate of missing or quitting the job, taking leave from the job, and getting ill, after controlling for the same confounders. Odds ratio, z-value, *p*-value, and 95% confidence interval were used to determine the association. A *p*-value of <0.05 with a 95% confidence interval (CI) was considered statistically significant.

## 3. Results

### 3.1. Sample Characteristics

The findings were part of the baseline survey of a cluster randomized controlled trial from the intervention arms, comprising a total of 867 pairs of senior citizens and their primary family caregivers. Of the 867 caregivers, 71 were identified as caregivers of older adults with depression (8.2%) and 796 as caregivers of older adults without depression (91.8%). The baseline demographic and clinical characteristics of the family caregivers are shown in Table 1. The mean age was 55.3 ± 13.8 years. Just over half of the caregivers were younger than 60 years of age (53.5%), more than one-third were 60–69 years old (35.0%), and more than 10% were older than 70 years (11.5%). The caregivers were more likely to be female than male (62.3% vs. 37.7%). Half (49.9%) of the caregivers were the main income supporters of the family, while 31.1% did not have a current job. About one-quarter (23.3%) of family caregivers were working as government office/company staff. As for the underlying diseases, hypertension was the most common disease (28.0%), followed by diabetes (9.3%) and hyperlipidemia (8.2%). When exploring the health behaviors, “no exercise” (21.0%), “current alcohol drinkers” (27.7%), and current smokers (8.8%) were reported (Table 1).

Bivariate analysis showed no difference between the two groups in age, gender, occupation, being the main income supporter of the family, underlying diseases, and exercise habits. However, the caregivers of older adults with depression were more likely to be single (31.0% vs. 20.2%) and less educated (45.1% vs. 31.9%) than the caregivers of older adults without depression.

### 3.2. Care Recipient’s GDS Total Scores

Of the 867 older adults, 71 older adults (8.2%) had GDS total scores of 5–15, indicating that they have probable depression. Among them, 50 older adults (5.8%) had scores of 5–8 suggesting mild depression, 13 (1.5%) had scores of 9–11 suggesting moderate depression, and 8 (0.9%) had scores of 12–15 suggesting severe depression.

### 3.3. Bivariate Analyses of Outcome Measures

For the outcome measures, bivariate analyses showed significant differences between the two groups both in the CBI total score (14.8 vs. 3.4) and in the EQ-VAS score (75.0 vs. 81.7), showing that caregivers of older adults with depression experienced lower health-related QOL and greater caregiver burden (Table 2). Regarding the five subscales of CBI, in all areas of time, development, health, emotion, and social subscale, the caregivers of older adults with depression felt a greater burden than the caregivers of older adults without depression (all *p*-values < 0.001). As for the five dimensions of EQ-5D-5L, there were significant differences in mobility (*p* < 0.001), usual activities (*p* = 0.001), and pain/discomfort (*p* < 0.001) dimensions, while there was no significant difference in self-care and anxiety/depression dimensions. While there was no significant difference in the rate of missing or quitting job, the caregivers of older adults with depression were more likely to take leave from their jobs (*p* = 0.001) and get ill (*p* < 0.001) than the caregivers of older adults without depression (Table 2).

### 3.4. Multivariable Analyses of Outcome Measures

Multivariable analyses showed a significant difference between the two groups in the CBI total score (Coefficient: 10.60, 95% confidence interval (CI): 8.60–12.60), showing that the caregivers of older adults with depression experienced greater caregiver burden (Table 3). Regarding the five subscales of CBI, in all areas of time (Coefficient: 4.36, 95% CI: 3.55–5.17), development (Coefficient: 1.53, 95% CI: 1.12–1.94), health (Coefficient: 1.81, 95% CI: 1.40–2.21), emotion (Coefficient: 1.42, 95% CI: 0.91–1.93), and social (Coefficient: 1.49, 95% CI: 1.04–1.93) subscales, the caregivers of older adults with depression felt a greater burden than the caregivers of older adults without depression (all *p*-values < 0.001). The multivariable analyses showed a significant difference between the two groups in the EQ-VAS score (Coefficient: −5.52, 95% CI: −8.41–−2.62), showing that the caregivers of older adults with depression experienced lower health-related QOL. As for the five dimensions of EQ-5D-5L, there were significant differences in mobility (Coefficient: 0.13, 95% CI: 0.05–0.21), usual activities (Coefficient: 0.16, 95% CI: 0.08–0.25), and pain/discomfort (Coefficient: 0.19, 95% CI: 0.07–0.31) dimensions, while there was no significant difference in self-care (Coefficient: 0.04, 95% CI: −0.05–0.12) and anxiety/depression dimensions (Coefficient: 0.06, 95% CI: −0.04–0.15). While there was no significant difference in the rate of missing or quitting job (adjusted odds ratio (Adj OR): 1.64, 95% CI: 0.59–4.51), the caregivers of older adults with depression were more likely to take leave from their jobs (Adj OR: 4.00, 95% CI: 1.73–9.24) and get ill (Adj OR: 7.26, 95% CI: 2.68–19.64) than the caregivers of older adults without depression (Table 4).

## 4. Discussion

This study compared the caregiving burden of caregivers of older adults with depression and without depression in Thailand. Even after adjusting for possible confounding factors, the caregiving burden of caregivers of older adults with depression was significantly higher in the CBI total score and all five subscales of the CBI. In addition, EQ-VAS was significantly lower among the caregivers of older adults with depression. As for the five dimensions of EQ-5D-5L, there were significant differences in mobility, usual activities, and pain/discomfort dimensions, while there was no significant difference in self-care and anxiety/depression dimensions. Additionally, although no significant difference was found in the likelihood of missing or quitting a job, it was found that the caregivers of older adults with depression were significantly more likely to take leave from their jobs and more likely to get ill.

In the present study, about 62% of the caregivers are women, which is equal to or higher than the previous studies [23,32,33]. This may reflect Thai culture, where women are the main caregivers for family members.

The mean age of the caregivers is 55.3 years, which is older than that of the other studies assessing the caregiving burden of caregivers of people with depression [22,34]. Many caregivers are in their productive age, and about half of them are the main income supporters, but caring for their families may impair their ability to work, sometimes resulting in being absent from work or losing work. Because some caregivers themselves are in their old age, their physical strength may be deteriorating, and they are more likely to feel the burden of caregiving. Some caregivers themselves have chronic conditions such as hypertension or diabetes, so they must balance their health with caring for their family.

The caregivers of older adults with depression are less likely to be married and more likely to be single or not currently married than the caregivers of older adults without depression. This result is consistent with a previous study [23]. Within a given family, single people are more likely to be the caregivers of older adults with depression, and some may even divorce as a result of caring for older adults with depression.

The caregivers of older adults with depression are likely to be less educated, which is specific to the present study [23,33]. This may reflect the fact that the prevalence of depression is closely associated with sociodemographic factors like education, occupation, and income [35]. Namely, older adults with low sociodemographic status are likely to become depressive, and their caregivers may be less educated. To make matters worse, caregivers with lower levels of education are less likely to respond well to the pressure of caregiving [36]. Care capacity building is important for them.

The caregivers of older adults with depression reported significantly higher caregiver burden for all five subscales of time, development, health, emotion, and social than the caregivers of older adults without depression. This result is consistent with the previous studies that reported that the burden of caregiving in depression is equal to or higher than caregiving in other mental or physical conditions [23,37,38]. This may be due to the unpredictable progression pathway of the disease, the demand for constant supervision, and the inappropriate behavior of the patient. Even if caregivers are providing adequate care and affection, patients may not understand the caregiver. Additionally, some caregivers suffer from stigma, resulting in blame, social isolation, and discrimination [39,40].

As for health-related QOL, the present research revealed that the caregivers of older adults with depression experienced lower overall QOL and QOL in the dimensions of mobility, usual activities, and pain/discomfort. This result seems consistent with the prior research suggesting that caregiving had bad effects on the caregivers’ physical conditions [33]. However, there was no significant difference in the dimensions of self-care and anxiety/depression. This result is different from the previous studies, which revealed that caregivers experienced a stronger influence on mental functioning than physical functioning [23,33]. So, significant differences in the overall QOL and QOL in some dimensions may not suggest the consequence of caregiving but imply the basic characteristics of caregivers of older adults with depression. Namely, caregiving for older adults with depression involves less physical care than caregiving for older adults with other physical illnesses, so caregivers of older adults with depression may include physically weaker caregivers. Older adults’ dependency assessed using Barthel’s Activity of Daily Living index is known to be associated with their health-related QOL, so this might have influenced the results [41]. Based on this idea, it can be said that the significant impact of caregiving for older adults with depression on health-related QOL was not found because there was no significant difference in the dimension of anxiety/depression. This might be because the symptoms of people with depression in the present study were mostly mild. The past researches have shown that the more severe the depression, the greater the burden on caregivers and the loss of QOL [34,42,43]. Additionally, because EQ-5D-5L is a measure that is often used in clinical settings, the use of other measures of health-related QOL might have revealed additional information.

Prior studies have shown that caregiving for adults with depression is associated with work impairment [23,44]. The results of the present study supported this fact. The possibility of missing or quitting a job was not significantly different, but the caregivers of older adults with depression were more likely to take leave from their jobs. This can be attributed to the time required for caregiving and mental problems such as depression and insomnia of caregivers [33]. Taking measures to make it easier for caregivers to take leave from their jobs can reduce their burden [45]. In the present study, the caregivers of older adults with depression were more likely to get ill, which is also consistent with past research that showed a higher rate of health resource utilization of caregivers of patients with depression [23,33]. It is important to take measures like respite care services to let caregivers take a rest from caregiving and seek medical consultations [45].

It is also important that this study was conducted in Thailand, a middle-income country. Much of the evidence on the caregiving burden of older adults has come from high-income countries, but population aging is progressing even in low- and middle-income countries [23,33]. Moreover, in such countries, facility-based care and respite care are not yet in place, so it is important to consider the caregiving burden of informal caregivers and ways to reduce it [46].

Interventions aimed at reducing the burden on caregivers may address which types of caregivers are likely to have higher levels of burden. A study on cancer patients and their caregivers found that when the cancer patients underestimated their caregivers’ caregiving burden, their caregivers perceived lower QOL and higher levels of depression and anxiety [47]. In dementia care, some studies suggested that caregivers who perceived positive aspects of caregiving and who felt that they were receiving sufficient social support were likely to feel less caregiving burden and had a lower risk of developing depression [21,48,49]. One study found that caregivers of people with late-life depression who believed that “depression is caused by the patient’s own personality or laziness” or “depressive symptoms are intentional and are done to manipulate people” were likely to perceive greater caregiving burden [50]. As mentioned above, caregivers with lower levels of education are less likely to respond well to the pressure of caregiving [36]. Given these facts, psychological support for caregivers is necessary, and it is also important to improve their care capacity by providing them with correct knowledge of depression and coping methods.

A study in Korea found that community-based dementia caregiver intervention was effective in reducing the caregiving burden of caregivers of dementia patients [51]. This intervention included education and training to improve knowledge of dementia and dementia-related services and systems and the ability to care for people with dementia. Furthermore, research in Thailand has shown that it is possible to reduce the caregiver burden by creating a system that comprehensively supports caregivers with a combination of formal and informal care, including care prevention, care capacity building, and community respite service [46]. Therefore, scaling up the evidence-based community-integrated intermediary care model in the district and subdistrict level will optimize the systematic response to the challenge of population ageing in Thailand [46].

### 4.1. Recommendations

The findings of this study revealed important policy implications. Less education and lack of knowledge of caregivers impose additional caregiving burdens, so care capacity building by providing them with correct knowledge of depression and coping methods is important. Caregivers of adults with depression experience emotional burdens, so psychological support is also needed. For this purpose, peer-support groups for caregivers may be effective [52]. They are likely to get ill and take leave from their job, so taking measures to make it easier for them to take leave from their jobs can reduce their burden [45]. Providing respite care service is also important to give caregivers chances to take good rest and enjoy their own lives. Some caregivers may lose income by taking leave or quitting jobs, so financial support can be helpful. It is necessary to reduce the burden on caregivers of older adults with depression by combining these measures, in order to sustain family centered long term care.

### 4.2. Strengths and Limitations

The strength of this study is that it includes a sufficient number of pairs of older adults and their caregivers and conducts a multivariable analysis that is adjusted for various potentially confounding sociodemographic characteristics. This study is not without limitations. It is a cross-sectional study that used baseline data of a cluster randomized control trial, so although it can explain the association between depression in older adults and the caregiving burden on caregivers, it cannot claim a causal relationship. Additionally, the classification of depression is based on people’s responses to a screening tool and is not medically confirmed. Moreover, this study did not take older adults’ ADL scores and did not analyze what kind of care they received. These factors could have been confounders of the association among older adults having depression and family caregivers’ caregiving burden, health-related QOL, physical condition, and employment. Additionally, future qualitative research for caregivers of older adults with depression would provide more information contributing to developing pragmatic and effective measures to relieve their burden. Finally, implementation research is required to determine the effectiveness of various measures.

## 5. Conclusions

This study suggests that the caregivers of older adults with depression have a greater caregiving burden, have low health-related QOL, and are more likely to get sick and take leave from their jobs. Thailand has to be ready for the forthcoming super-aged society by establishing a sustainable long-term care system and addressing the problem of the high burden on caregivers. It is necessary to take measures to reduce the burden on caregivers of older adults with depression. There should be a holistic approach, including care capacity building, psychological support, respite care service, financial support, and other health promotion activities. Future research to analyze the detailed pathway of greater caregiver burden and to determine the effectiveness of various measures to relieve caregiver burden is required.

## Figures and Tables

**Table 1 medicina-61-00050-t001:** Demographic and clinical characteristics of the family caregivers.

Demographic Characteristics	Caregivers of Older Adults Total (*n* = 867)	Caregivers of Older Adults with Depression (*n* = 71)	Caregivers of Older Adults Without Depression (*n* = 796)	*p*-Value
	*n* (%)	*n* (%)	*n* (%)	
Age (mean ± SD)	55.3 ± 13.8	55.2 ± 13.9	55.3 ± 13.7	0.940
<30 years	36 (4.2)	1 (1.4)	35 (4.4)	
30–39 years	100 (11.5)	12 (16.9)	88 (11.1)	
40–49 years	138 (15.9)	12 (16.9)	126 (15.8)	
50–59 years	190 (21.9)	17 (23.9)	173 (21.7)	
60–69 years	303 (35.0)	20 (28.2)	283 (35.6)	
≥70 years	100 (11.5)	9 (12.7)	91 (11.4)	
Gender				
Male	327 (37.7)	28 (39.4)	299 (37.6)	0.755
Female	540 (62.3)	43 (60.6)	497 (62.4)	
Marital Status				
Single	183 (21.1)	22 (31.0)	161 (20.2)	0.003
Married	611 (70.5)	38 (53.5)	573 (72.0)	
Not currently married (separated, divorced, widowed)	73 (8.4)	11 (15.5)	62 (7.8)	
Education				
Primary school	286 (33.0)	32 (45.1)	254 (31.9)	0.024
Secondary school and above	581 (67.0)	39 (54.9)	542 (68.1)	
Occupation				
No job	270 (31.1)	23 (32.4)	247 (31.0)	0.273
Own business	253 (29.2)	17 (23.9)	236 (29.7)	
Company/office staff	202 (23.3)	14 (19.7)	188 (23.6)	
Daily labor	142 (16.4)	17 (23.9)	125 (15.7)	
Main Income Supporter of Family				
No	434 (50.1)	29 (40.9)	405 (50.9)	0.105
Yes	433 (49.9)	42 (59.2)	391 (49.1)	
Underlying Diseases				
Hypertension				
No	624 (72.0)	54 (76.1)	570 (71.2)	0.424
Yes	243 (28.0)	17 (23.9)	226 (28.4)	
Diabetes				
No	786 (90.7)	63 (88.8)	723 (90.8)	0.561
Yes	81 (9.3)	8 (11.3)	73 (9.2)	
Hyperlipidemia				
No	796 (91.8)	65 (91.6)	731 (91.8)	0.933
Yes	71 (8.2)	6 (8.4)	65 (8.2)	
Current Smoker				
No	791 (91.2)	65 (91.6)	726 (91.2)	0.922
Yes	76 (8.8)	6 (8.4)	70 (8.8)	
Current Alcohol Drinker				
No	627 (72.3)	48 (67.6)	579 (72.7)	0.354
Yes	240 (27.7)	23 (32.4)	217 (27.3)	
Exercise				
No exercise	182 (21.0)	22 (31.0)	160 (20.1)	0.087
Exercise but not regular	569 (65.6)	42 (59.2)	527 (66.2)	
Regular exercise	116 (13.4)	7 (9.9)	109 (13.7)	

Notes: Student *t*-tests were used for continuous variables, and Pearson’s Chi-Square tests were used for categorical variables.

**Table 2 medicina-61-00050-t002:** Bivariate analyses of outcome measures among the caregivers of older adults with and without depression.

Outcome Measures	Caregivers of Older Adults Total (*n* = 867)	Caregivers of Older Adults with Depression (*n* = 71)	Caregivers of Older Adults Without Depression (*n* = 796)	*p*-Value
	*n* (%)	*n* (%)	*n* (%)	
CBI Total Score	4.37 ± 9.1	14.79 ± 14.29	3.44 ± 7.82	<0.001
<24	815 (94.0)	52 (73.2)	763 (95.9)	<0.001
≥24	52 (6.0)	19 (26.8)	33 (4.2)	
CBI Time Score	1.52 ± 3.64	5.89 ± 6.60	1.13 ± 2.96	<0.001
0	641 (73.9)	25 (35.2)	616 (77.4)	<0.001
≥1	226 (26.1)	46 (64.8)	180 (22.6)	
CBI Develop Score	0.60 ± 1.78	2.11 ± 3.17	0.47 ± 1.53	<0.001
0	735 (84.8)	39 (54.9)	696 (87.4)	<0.001
≥1	132 (15.2)	32 (45.1)	100 (12.6)	
CBI Health Score	0.68 ± 1.76	2.39 ± 3.44	0.53 ± 1.43	<0.001
0	680 (78.4)	31 (43.7)	649 (81.5)	<0.001
≥1	187 (21.6)	40 (56.3)	147 (18.5)	
CBI Emotion Score	0.83 ± 2.19	2.24 ± 3.54	0.71 ± 1.99	<0.001
0	702 (81.0)	39 (54.9)	663 (83.3)	<0.001
≥1	165 (19.0)	32 (45.1)	133 (16.7)	
CBI Social Score	0.73 ± 1.92	2.15 ± 3.04	0.60 ± 1.73	<0.001
0	720 (83.0)	38 (53.5)	682 (85.7)	<0.001
≥1	147 (17.0)	33 (46.5)	114 (14.3)	
EQ-VAS (mean ± SD)	81.1 ± 12.7	75.0 ± 14.7	81.7 ± 12.3	<0.001
EQ-Mobility (mean ± SD)	1.09 ± 0.33	1.23 ± 0.45	1.08 ± 0.32	<0.001
No problem	795 (91.7)	56 (78.9)	739 (92.8)	<0.001
More than slight problem	72 (8.3)	15 (21.1)	57 (7.2)	
EQ-Self-Care (mean ± SD)	1.09 ± 0.34	1.13 ± 0.38	1.09 ± 0.34	0.231
No problem	801 (92.4)	63 (88.7)	738 (92.7)	0.225
More than slight problem	66 (7.6)	8 (11.3)	58 (7.3)	
EQ-Usual Activities (mean ± SD)	1.11 ± 0.36	1.28 ± 0.66	1.10 ± 0.32	0.001
No problem	779 (89.9)	56 (78.9)	723 (90.8)	0.001
More than slight problem	88 (10.1)	15 (21.1)	73 (9.2)	
EQ-Pain/Discomfort (mean ± SD)	1.25 ± 0.51	1.45 ± 0.60	1.23 ± 0.49	<0.001
No problem	674 (77.7)	43 (60.6)	631 (79.3)	<0.001
More than slight problem	193 (22.3)	28 (39.4)	165 (20.7)	
EQ-Anxiety/Depression (mean ± SD)	1.13 ± 0.39	1.20 ± 0.47	1.13 ± 0.38	0.151
No problem	765 (88.2)	59 (83.1)	706 (88.7)	0.161
More than slight problem	102 (11.8)	12 (16.9)	90 (11.3)	
Miss or Quit Job				
No	818 (94.3)	65 (91.6)	753 (94.6)	0.281
Yes	49 (5.7)	6 (8.5)	43 (5.4)	
Take Leave from Job				
No	827 (95.4)	61 (85.9)	766 (96.2)	0.001
Yes	40 (4.6)	10 (14.1)	30 (3.8)	
Get ill				
No	843 (97.2)	63 (88.7)	780 (98.0)	<0.001
Yes	24 (2.8)	8 (11.3)	16 (2.0)	

Notes: Wilcoxon signed-rank tests were used for continuous variables, and Fisher’s exact tests were used for categorical variables. Abbreviations: CBI, caregiver burden inventory scale; EQ, European Quality of Life scale; VAS, visual analogue scale; SD, standard deviation.

**Table 3 medicina-61-00050-t003:** Effects of the presence of depression in the care recipients on for caregivers’ burden and their quality of life, multiple regression analysis results.

Outcome Measures	Coefficient	t-Value	*p*-Value	95% CI	
				Lower	Upper
CBI total score	10.60	10.41	<0.001	8.60	12.60
CBI time score	4.36	10.53	<0.001	3.55	5.17
CBI develop score	1.53	7.30	<0.001	1.12	1.94
CBI health score	1.81	8.74	<0.001	1.40	2.21
CBI emotion score	1.42	5.44	<0.001	0.91	1.93
CBI social score	1.49	6.59	<0.001	1.04	1.93
EQ-VAS	−5.52	−3.74	<0.001	−8.41	−2.62
EQ-Mobility	0.13	3.27	0.001	0.05	0.21
EQ-Self-Care	0.04	0.84	0.400	−0.05	0.12
EQ-Usual Activities	0.16	3.69	<0.001	0.08	0.25
EQ-Pain/Discomfort	0.19	3.09	0.002	0.07	0.31
EQ-Anxiety/Depression	0.06	1.18	0.238	−0.04	0.15

Notes: All these measures are analyzed as continuous variables, and multiple regression analysis was applied, adjusting for age, sex, marital status, smoking, alcohol drinking, exercise habits, health status, diabetes mellitus, education, occupation, and being the main income supporter of the family. Abbreviations: CBI, caregiver burden inventory scale.

**Table 4 medicina-61-00050-t004:** Effects of the presence of depression in the care recipients on caregivers, multiple logistic regression result.

Outcome Measures	Odds Ratio	z-Value	*p*-Value	95% CI	
				Lower	Upper
Miss or quit the job	1.64	0.95	0.343	0.59	4.51
Take leave from the job	4.00	3.24	0.001	1.73	9.24
Get ill	7.26	3.90	<0.001	2.68	19.64

Notes: Logistic regression analysis was applied, adjusting for age, sex, marital status, smoking, alcohol drinking, exercise habits, health status, diabetes mellitus, education, occupation, and being the main income supporter of the family. Abbreviations: CI, confidence interval.

## Data Availability

The data presented in this study are available on request from the corresponding author due to privacy reasons.

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
