# Peer review of "Substantial Impact of Later-Life Depression Among Community Older Adults on the Family Caregivers’ Burden in the Home Care Setting of Chiang Mai, Northern Thailand"

_medicina, 2025, doi:10.3390/medicina61010050_

Round 1
Reviewer 1 Report
Comments and Suggestions for Authors
Thank you for the opportunity to review the manuscript ID: medicina-3375178. This manuscript aimed to investigate the impact of later life depression among community older adults upon the family caregivers' burden in Thailand.
This paper deals with a current and very important topic, which is shown in a very detailed manner in the Introduction section. Consequently, the goal of the work is clearly defined.
Comments refer to other sections of this manuscript.
Comments:
A dozen references were published more than 20 years ago. Correct this.
Line 101: Add a new paragraph in which the `Study design` applied in this paper should be defined.
Lines 148-160: State the psychometric properties of the Caregiver Burden Inventory in Thai. Cite the appropriate reference.
Line 160: For the Caregiver Burden Inventory, state the scoring method used in this study, citing the appropriate reference.
Lines 162-170: State the psychometric properties of the EQ-5D-5L scale in Thai. Cite the appropriate reference.
Lines 171-172: Cite the appropriate reference for the information given in this sentence.
Lines 188-196: List the names of all association measures that were used to determine the association between the exposure variable and health-related QOL and care burden measures for the family caregivers' in this manuscript.
Lines 221-222: Under Table 1, indicate which statistical test was applied to determine `p`.
Lines 243-245: Under Table 2, state which statistical test was applied to determine `p`.
Lines 247-263 and Lines 269-273: What statistical procedure was used to analyze the data described in this text and shown in Table 3? Write under Table 3. Correct this throughout the paper.
Lines 410-415: The limitations of this manuscript are insufficiently discussed. Correct this. Also, indicate possibilities for overcoming the limitations of this work.
Reviewer 2 Report
Comments and Suggestions for Authors
Many thanks for the opportunity to review a scientific article Substantial impact of later life depression among community 2 older adults upon the family caregivers’ burden, in the home 3 care setting of Chiang Mai, Northern Thailand. After review analysis, I am sending the following comments on improving the scientific article. Each section of the article has been separately described and evaluated.
References. In the citations section, please complete the following items in the literature (no pages, no numbers) -1,2,7,9,11,13,15, 39, 45. Also, shouldn't item number 29 from 1989 be changed to a more recent source.
Abstract. The abstract is very factual outlining all parts of the scientific paper however, please expand the practical application in the abstract.
Key words. Please reduce the keywords in the article. Is the word Asia necessary in keywords, rather not?
Introduction. The introduction to the topic of the scientific paper is insufficient, the authors should present the problem of depression more from the scientific side and not from the statistical side (what is the percentage of incidence). This is missing from the introduction. Reading the introduction, it is not clear what the specific aim of the article is.
Material and methods. The number of people surveyed is impressive, but this section lacks detailed information on what the selection of survey participants was like.
The number of people surveyed is impressive, but this section lacks detailed information on what the selection of survey participants was like.
Results. The results of the study lack extended information with a description of the abbreviations used under each table. (for example p value).
Discussion. In the discussion, the authors compare their findings with other scientific publications which shows the scientific quality.
Conclusions. In the conclusions, the authors should add a practical conclusion and expand on the conclusions drawn from the research results obtained. In addition, the authors should add information about future works.
All manuscripts. Throughout the scientific article, the authors should correct errors such as unnecessary space, double spacing, if it is the first time writing an abbreviation then the whole name should be added as well.
